# Effect of Mn Content on the Toughness and Plasticity of Hot-Rolled High-Carbon Medium Manganese Steel

**DOI:** 10.3390/ma16062299

**Published:** 2023-03-13

**Authors:** Menghu Wang, Xiaokai Liang, Wubin Ren, Shuai Tong, Xinjun Sun

**Affiliations:** Central Iron & Steel Research Institute, Haidian District, Beijing 100082, China

**Keywords:** high-carbon manganese steel, plasticity and toughness, sharp hard phase, transverse cracks, deformation mechanism

## Abstract

The tensile and impact deformation behavior of three different Mn content test steels, xMn-1.0C-0.25V-1.5Cr-0.3Mo (5, 8 and 13 wt%), were investigated using mechanical properties testing, SEM-EBSD and TEM. The elongation and −20 °C impact energy of the three types of Mn content test steels increased as the Mn content increased. The room temperature tensile elongation was 9%, 23% and 81%, and the −20 °C impact energy was 9 J, 99 J and 241 J, respectively. The fracture morphologies of 5 Mn and 8 Mn were found to be cleavage fractures with secondary cracks and micro-voids. The 13 Mn fracture morphology was a plastic fracture with many coarse dimples. Transverse cracks perpendicular to the tensile direction occurred on the surface of the gauge area of 5 Mn and 8 Mn tensile specimens, reducing plasticity dramatically. This was mainly related to the martensitic transformation produced by stress. We characterized the martensite near the tensile fracture and speculated the main mode of crack propagation. Furthermore, a little amount of sharp-shaped BCC phase was found in the 5 Mn, which was determined to be a hard phase relative to the austenite matrix by nanoindentation test. These steels have stacking fault energies ranging from ~15 to ~29 mJ/m^2^ with increasing Mn content 13 Mn has high stacking fault energy (SFE) and austenite stability. Twin-induced plasticity (TWIP) was the deformation mechanism.

## 1. Introduction

High manganese steel, also known as “hadfield steel,” was created in 1882 by the British R.A. Hadfied. This form of steel typically has a high carbon content of 0.90% to 1.50% and contains 10% to 15% manganese [1]. This material has a high surface hardening ability, while the core retains its original high plasticity and toughness. Its wear resistance is significantly greater than that of other steel materials under high-impact loads. However, under moderate- or low-impact loading, high manganese steel exhibits low yield strength and poor work-hardening ability [2]. In order to solve these problems, the solid solution and precipitation strengthening methods of adding Cr, Mo, Ti and V elements to the steel were adopted [3,4,5], and the Mn element was reduced to obtain better work-hardening ability of Hadfield steel under different impact loads [6,7,8,9]. Through the above efforts, medium manganese steel based on Hadfield steel can be obtained and widely used in heavy industrial machinery.

However, the research on the effect of Mn content on the microstructural characteristics and the ensuing mechanical properties of high-carbon medium manganese steel is still insufficient. Reducing the Mn concentration of traditional high manganese steel to the range of medium manganese steel will weaken the stability of austenite and reduce SFE [10]. The SFE of medium manganese steels can be used to predict the deformation mechanism. As the SFE increased, the deformation process shifted from transformation-induced plasticity (TRIP) at low SFE (18 mJ/m^2^) to TWIP at high SFE (from 18 mJ/m^2^ to 40 mJ/m^2^) [11]. However, not all martensitic transformations are beneficial to plasticity and toughness. Deformation-induced martensite transformation (DIMT) often deteriorates ductile toughness in austenitic steel [12]. Ishigami et al. [13] studied the effect of Mn addition on the plasticity and toughness of high-carbon medium manganese steel. The Mn addition improved the tensile properties in the steel, particularly the strength and local deformability balance. This is mainly because the addition of Mn improves the thermal stability of lamellar cementite. Huang et al. [14] studied the relationship between the morphology of martensite formed by the DIMT effect and the mode of crack propagation on impact toughness in austenitic steels. It is proved that the crack propagation is closely related to martensite. In recent years, some scholars [15,16,17] have studied it. There is still a lack of analysis of the fine characterization and propagation mechanism of cracks. At present, there is no summary of the microstructural characteristics and the ensuing mechanical properties of high-carbon medium manganese steel.

In the present study, the effects of Mn content on the deformation mechanism, impact and tensile failure behavior of high-carbon medium manganese austenitic steels with three different Mn contents were investigated. The crack nucleation and growth mechanisms of this research can be used as a basis for further study of the low plasticity and toughness of high-carbon medium manganese steel.

## 2. Experimental Materials and Procedure

Table 1 lists three types of experimental steel chemistries, which are referred to as “5 Mn,” “8 Mn" and “13 Mn.” Thermocala calculation software was used to estimate that the non-equilibrium austenite phase region began to precipitate cementite below 895 °C. The final rolling temperature was set at 900 °C to achieve full austenite and minimize the influence of carbides.

Smelting in a 100 kg vacuum induction furnace, forging billets of 120 mm width by 60 mm thickness and rolling after forging, each number at 1150 °C for 3 h, rolling out of the furnace to ensure that the final pass is above 900 °C, water quenching to room temperature.

The tensile bar sample φ10-M16 was obtained from a hot-rolled plate parallel to the rolling direction and stretched at room temperature at a set rate of 5 × 10^−3^ s^−1^ using the WE-300 tensile machine. At −20 °C, the Charpy impact test was performed on the sub-size Charpy V-notch specimen. The V-notch specimen has an opening direction that is parallel to the rolling direction.

The sample was taken from a hot-rolled plate, and the planes of the rolling and normal directions were polished before corrosion with 4% nitric acid alcohol. The optical microscope (OM; Olympus BX41) and scanning electron microscopy (SEM; Quanta 650) were used for electron backscatter diffraction characterization (EBSD; Oxford 7900), the step size was set to be 0.3 μm and the samples were prepared using normal metallographic processes. However, a longer polishing time is required to effectively remove the surface stress layer and determine the phase structure using the image analysis program (Channel 5). The transmission electron microscopy (TEM) sample was taken from the gauge part of the tensile specimen near the fracture, and the plane was perpendicular to the tensile direction. The cut sheets were mechanically ground to a thickness of 30 m and ion-thinned to perforation using Gatan ion polishing equipment. Finally, TEM testing was carried out in JEM-2100F. X-ray diffraction (XRD) was used to quantify and qualitatively examine the dislocation density and phase type in the material. For steel materials, the Co target test is usually used.

## 3. Results and Discussion

### 3.1. Effect of Mn on Mechanical Properties

The SFE of three test steels with different Mn contents was calculated using the research of Saeed-Akber and Decooman et al. [11,18,19] (Table 2). The SFE in austenitic steel not only represents the minimal driving force required for twinning, but also the critical stress of the austenite-phase martensitic transition [20,21]. The energy of stacking faults was mainly related to the composition of medium and high manganese steel [22,23]. Since Mn can make austenite more stable, it can be assumed that as Mn content rises, SFE also does so, and the tendency of stress-induced phase change gradually decreases. According to the range of SFE, it is speculated that the three test steels should have different deformation mechanisms.

According to the tensile and impact test results shown in Table 2, the elongation and impact toughness of the three test steels increased steadily with an increasing Mn content, with the elongation of 13 Mn increasing by 352% compared to 8 Mn, and the impact toughness increasing by 248%. The 5 Mn, in contrast, has extremely low plasticity and toughness.

The true stress-strain curves of the three sample steels are shown in Figure 1, and the three curves are approximately proportional to the increasing trend, indicating that the straining process of the three sample steels has maintained a certain work-hardening rate. With the exception of 13 Mn, the strain hardening rate began to fall before the true strain reached 0.5. The low plasticity of 5 Mn and 8 Mn results in incomplete work-hardening and early fracture behavior. The serrated shape can be observed in all three curves at the same time, which is related to the dynamic strain aging (DSA) effect [24]. It is a phenomenon in which C-Mn atoms pile up to block dislocations, causing stress to pile up and then unscrew. This process caused the flow stress to change constantly. The jagged fluctuation position of 5 Mn test steel with low plasticity fluctuates greatly in the local enlarged figure inside the dotted frame of Figure 1, indicating that the stress required for depinning after dislocation accumulation in C-Mn atomic clusters is large, and the pinning effect is obvious, and the local dislocation density is high, causing local stress concentration. According to the aforementioned, the lower stacking fault energy and higher stress-induced phase transformation tendency of 5 Mn will cause a significant amount of martensite formation. It is worth noticing that while all three curves include serrated features, their serrated types differ. The serrated shapes in medium manganese steel are divided into the following three types: A, B and C [25,26,27]. Serrations of 5 Mn and 8 Mn are C-type (sudden stress decrease, then normal stress level rises quickly [28,29]), whereas serrations of 13 Mn transition from A-type (stress peak arises between two consecutive arc curves) to C-type as strain increases. The main reason for this is that when the Mn content is low, the number of C-Mn atomic clusters is low, the C interstitial atoms are continuously pinned and depinned in a free state [28]. When Mn content is high, a large number of C-Mn atom pairs and stacking defects stack to increase work-hardening ability [28,30]. Furthermore, the higher the Mn content, the higher the stacking fault energy, and the smaller the stacking fault width, the lower the work-hardening ability once the real stress reaches its peak. The increased Mn content of 13 Mn may be the primary cause of type A serrated DSA weakness. The serrated morphology of 5 Mn and 8 Mn differs from that of 13 Mn, indicating that their main deformation mechanism differs from that of 13 Mn, resulting in different plasticity.

The toughness of the three test steels also varied significantly (Table 2). The impact energy at −20 °C normally increased as the Mn content increased. Due to the poor plasticity of the material, it can be concluded that the low-temperature toughness of 5 Mn was not caused by the ductile-brittle transition temperature. Different Mn content influences fracture morphology (Figure 2). The 5 Mn fracture was mainly brittle; the 8 Mn fracture included micro-pores and some tearing ridges and was generally quasi-cleavage in character. A high number of large and deep dimples were observed on the 13 Mn fracture, along with extended tearing edges, indicating a plastic fracture. The 5 Mn, unlike 8 Mn and 13 Mn, exhibited a high number of secondary cracks. The 8 Mn included micro-pores, but the proportion of secondary cracks is modest, while 13 Mn had no micro-cracks or pores at all. As a result, the formation of secondary cracks revealed that there were mechanisms causing crack propagation in the 5 Mn strain process.

### 3.2. Microstructure Difference of Mn Content

The three test steels were quenched and rolled using essentially the same processes, and EBSD detected essentially the same grain sizes in all three (Figure 3). The stability of austenite in the test steel increases with an increase in Mn content. However, after hot rolling and water quenching, the 5 Mn test steel with less Mn content basically formed full austenite (Figure 3a), which was also confirmed by XRD (Figure 4a) before deformation. In the 5 Mn matrix, there were sharp (red part of Figure 5c,d) phases, which were typically about a dozen microns in size and are mainly distributed near the grain boundaries. It may be due to the weak austenite stability of 5 Mn test steel [31], which will cause some element segregation and the formation of new phases. These sharp phases were defined as BCC structures using the processed phase distribution map (Figure 5c,d). SEM and energy dispersive spectroscopy (EDS) elemental analysis were used to identify the sharp phase. Figure 6 shows the scanning outcomes. The carbon content of the sharp phase was lower than that of the surrounding structure, but it is much higher than the 1 wt.% carbon content of the test steel. Because of the concentration of the C element in this region, dislocation movement will be hindered by an increase in the C-Mn atomic cluster content, and as a result, the local stress in this region will rise as a result of the straining process. A nanoindenter was also used to test the hardness of the BCC phase (as shown in Figure 7). The BCC phase had a higher hardness than the matrix austenite. Figure 5e,f shows KAM maps representative of the variation in microstructural strain in 5 Mn, with a minimum value of 0 (dark blue) and a maximum value of 3.5 (dark red). The BCC phase is in the red dotted circle. It can be found that these BCC phases are greenish, so the dislocation density is higher than that of the nearby structure. Because the martensite formed by quenching austenite is a non-diffusion phase transformation, the transformed martensite has high dislocation density. The tip of the sharp, hard phase will doubtlessly develop into a potential crack source during the straining process due to the localized stress concentration around the phase, which will reduce the plasticity and toughness. The phase content is estimated, and the total amount of this phase is limited. However, the total amount of this phase was small, and the phase content was estimated to be less than 5%. As a result, the sharp-shape hard phase had a very limited effect on plasticity and toughness. 

Figure 8 shows the tensile fracture of the three test steels. It can be shown that, whereas 13 Mn exhibited ductile fracture morphology, 5 Mn and 8 Mn exhibited brittle fracture morphology. It can be illustrated that the deformation mechanisms of 5 Mn and 8 Mn were essentially the same, but that 13 Mn was completely different. Moreover, the difference between 5 Mn and 8 Mn was mostly the number of secondary cracks, showing that crack symptoms lessened as Mn content increased. In the gauge part of the tensile specimen, there were some cracks perpendicular to the tensile direction (Figure 9). Transverse crack formation and propagation will directly reduce plasticity. In the specimen steel with various Mn contents, there were different numbers of cracks. It can be seen that 5 Mn and 8 Mn both have transverse cracks (perpendicular to the tensile direction) in the gauge section nearer the fracture, whereas 13 Mn only had a few microcracks. The entire gauge area along the tensile direction had transverse cracks of 5 Mn. When compared to 5 Mn, the distribution area of the transverse cracks in 8 Mn was much smaller and only occurs near the fracture. Transverse cracks in 13 Mn seemed to not typically exist, but the surface of the entire gauge zone became rough. This was primarily due to the TWIP effect, which distorted the twin grains and caused the floating convexity of the step-like structure grains to form. This resulted in the “rough” phenomenon on the sample’s surface without weakening the plasticity [32]. From Figure 9, it was clear that 5 Mn not only produced a greater number of transverse cracks but also that these cracks were distributed evenly throughout the tensile specimen. The metallographic image in Figure 10 shows that some transverse cracks can have depths of more than 0.5 mm, proving that the 5 Mn cracks are more likely to form and expand under tensile stress than cracks on other samples. It was found that the crack initiation position of the edge crack were both inside the grain and near the grain boundary, implying that the crack is primarily caused by surface geometry rather than structural defects. It is presumed that the tensile stress concentration becomes the crack initiation point when the geometry of the surface depression is approximately perpendicular to the tensile direction. Near the break of the “A zone” of the tensile fracture (Figure 10a,b), there are a few microcracks and microholes existing at the same time. It is determined through analysis that α′-martensite was formed close to these microcracks and microholes when combined with the phase distribution map. These tiny cracks spread away from the fracture in a direction perpendicular to the tensile direction. The severe strain close to the fracture is mainly responsible for the microcracks and microvoids in these samples. Table 3 summarizes the identified phases with their crystal structure and associated lattice parameters. The XRD patterns line near the fracture indicates that the phase transformation occurs after deformation, forming α′-martensite, in accordance with the stacking fault energy of 5 Mn test steel, which is 15 mJ/m^2^ (TWIP effect is difficult to occur) (as shown in Figure 4a dotted line). Figure 10c,d illustrates the α′-martensite close to the tensile fracture, and Figure 10g,h show ‘KAM’ map neartransverse cracks. It is evident that the amount of “martensite” produced decreases with increasing distance from the fracture, demonstrating that martensite is produced primarily as a result of strain. After stress concentration, a phase transition occurs. It is worth noting that the formation of martensite did not cause a decrease in local dislocation density. The local high-density dislocation around the “martensite” led to the continuous propagation of cracks.

Simple explanations for the TRIP effect of medium manganese austenitic steel can be found in [33] as follows: (1) When local deformation is at its peak, martensitic transformation tends to take place, increasing local strength and making it difficult for local crystals to continue deforming. Delaying the necking, the deformation propagates to the area devoid of phase transition. (2) Martensitic transformation has the ability to alleviate local internal stress, which prevents crack propagation. (3) There is a coherent relationship between the austenite matrix and the martensitic transformation; the interface energy is higher and can also prevent crack propagation. As a result, it exhibits high elongation. Due to the low stacking fault energy of 5 Mn material, partial dislocations are more widely separated or extended in the initial tensile strain process than in high-level fault energy materials. Partial dislocations are more widely separated or expanded in high SFE materials [34,35,36]. In the case of γ—α′-transformation in Austenitic steels, the dislocation pile-ups become sites of α′-martensite nucleation. Because the 5 Mn material has a low SFE, ε-martensite formed more easily than α′-martensite during the initial tensile strain process and then rapidly changed to α′-martensite [4,7,9,37]. Large amounts of formed α′-martensite cause a volume change that leads to microcracks and microvoids. In comparison to 8 Mn, 5 Mn had more ε-martensite to α′-martensite due to the difference in austenite stability. Different from the strain-induced martensite of isolated fine blocks, the α′-martensite has a large and continuous martensite block, which is beneficial to crack propagation and leads to premature fracture, thus limiting the tensile ductility [14,16]. If there is transformed martensite in the propagation direction, the microcrack will continue to expand perpendicular to the tensile direction. Finally, the difference in plasticity and toughness between 5 Mn and 8 Mn was caused. No phase transformation was observed on the peak line of 8 Mn before and after tensile deformation, probably because the content of α′-martensite was not enough. There was no phase transformation before and after the tensile deformation of 13 Mn test steel, and the diffraction peak after tensile deformation was obviously widened, indicating that the degree of grain distortion increases, and the dislocation density increases significantly. According to Figure 4c, calculating the dislocation density before and after stretching [38] revealed that the dislocation density before stretching increased by two orders of magnitude, from 1.72 × 10^9^ cm^−2^ to 6.50 × 10^11^ cm^−2^. This is mainly due to the fact that 13 Mn, as a typical high-level dislocation TWIP steel, produces a large number of mechanical twins after stretching to refined grains, which improves plasticity and toughness (as shown in Figure 11). Figure 11a–c shows bright field images and diffraction spots of a cross-section structure near a tensile fracture of 13 Mn steel, respectively. Figure 11a,b shows that there are many parallel and cross twins in the microstructure near the fracture of high manganese steel after fracture, with twin thicknesses ranging from 20 to 200 nm (as shown by the arrows at T1 and secondary twins). At the same time, there are secondary twins with different orientations and parallel arrangements in the twins, with a thickness of about 10 nm (as shown in the arrows in the figure). Furthermore, a large number of dislocation pile-ups and dislocation cell structures were observed at the interface between the matrix and the twin boundary, as well as between the twins (as shown in Figure 11b). When the stress reaches a certain level, a small secondary twin forms inside the primary twin, blocking dislocation slip. The primary twins formed during the plastic deformation of high-carbon, high-manganese steel are expected to grow further. A secondary twin with a thinner layer will form inside the main twin with a thicker layer as the strain increases to a certain point. The main mechanism of the TWIP effect [39,40,41] is the significant increase in dislocation density that results from the interaction of dislocations, primary twins and secondary twins, which can significantly improve the strain-hardening behavior of high-carbon, high-manganese steel.

## 4. Conclusions

(1) The tensile elongation at break and −20 °C impact toughness of the three test steels were 9% and 9 J, 23% and 99 J, and 81% and 241 J, respectively, as Mn content increased. In the impact test, the brittle fracture mechanism of 5, 8 Mn differed from that of 13 Mn, which exhibited ductile fracture. A large number of transverse cracks were distributed on 5 Mn in the gauge area of the sample after the tensile test sample fractures. The 8 Mn had transverse cracks only near the fracture, and 13 Mn had almost no cracks. The grains twisted and created a step-like structure, which caused the “wrinkle” phenomenon on the surface of the 13 Mn tensile sample. Because 5 Mn austenite is unstable, a sharp, hard BCC phase formed in the 5 Mn base material. The phase content of the BCC phase is less than 5%, and the effect on plasticity and toughness was quite limited.

(2) The stress-induced phase transformation produces martensite in low stacking fault energy (between 5 and 8 Mn) due to the concentration of stress near the tensile and impact fractures. The α′-martensite not only created microcracks and microvoids, but it also sped up crack propagation. In the microstructure of the 13 Mn steel, there were numerous deformation twins that range in thickness from 20 to 200 nm. The typical TWIP effect increased plasticity to 81% and impact toughness to 241 J. The low plasticity and toughness of 5, 8 Mn in high-carbon, medium-manganese and full austenite steel were due to austenite instability and the martensitic transformation induced by deformation, which was both helpful to crack generation and propagation. 

(3) The above experimental results show that there is no TRIP effect in the martensitic transformation process of all-austenitic steel. The transformed martensite will increase a certain strength but seriously deteriorate the plasticity. The crack propagation rate is too fast to increase the strength, which needs more theoretical calculations to verify. Inspired by this study, for wear-resistant austenitic medium manganese steel, the method of improving plasticity and toughness in the future can improve the stability of austenite by adding other alloying elements and trying to avoid the transformation of martensite; or through some heat treatment methods to change the stress-induced martensite of large and continuous martensite block morphology. This can avoid premature fracture caused by crack propagation.

## Figures and Tables

**Figure 1 materials-16-02299-f001:**
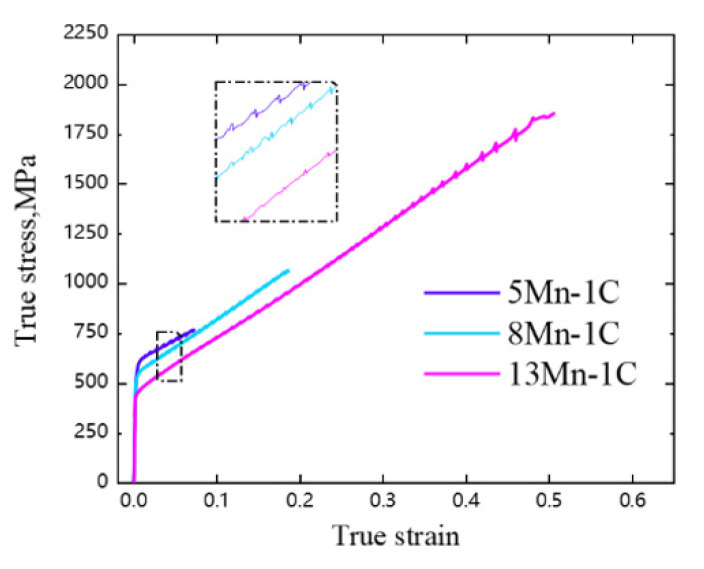
True stress-strain curves of three tested steels.

**Figure 2 materials-16-02299-f002:**
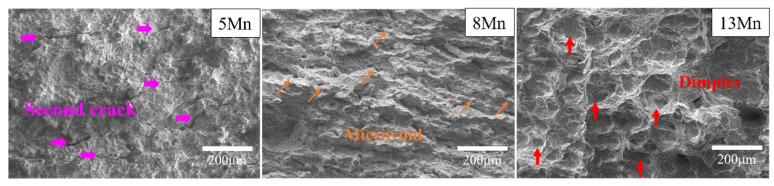
Impact fracture morphology of different Mn content alloy at −20 °C.

**Figure 3 materials-16-02299-f003:**
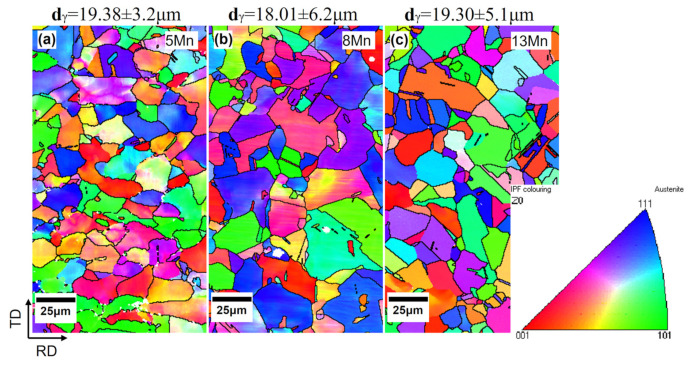
EBSD inverse pole figure (IPF) maps of the (**a**) 5 Mn, (**b**) 8 Mn and (**c**) 13 Mn steels. All the steels are composed of austenite grains of 18–20 μm in size.

**Figure 4 materials-16-02299-f004:**
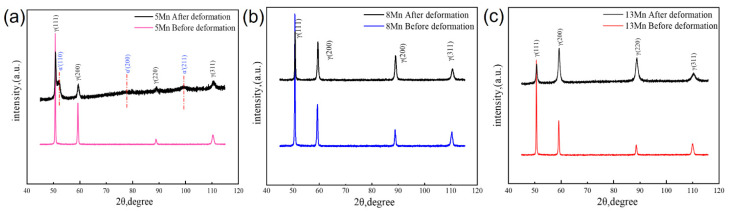
XRD before and after deformation (**a**) BCC phase formed after deformation of 5 Mn; (**b**,**c**) 8, 13 Mn half width of diffraction peak increases, dislocation density increases.

**Figure 5 materials-16-02299-f005:**
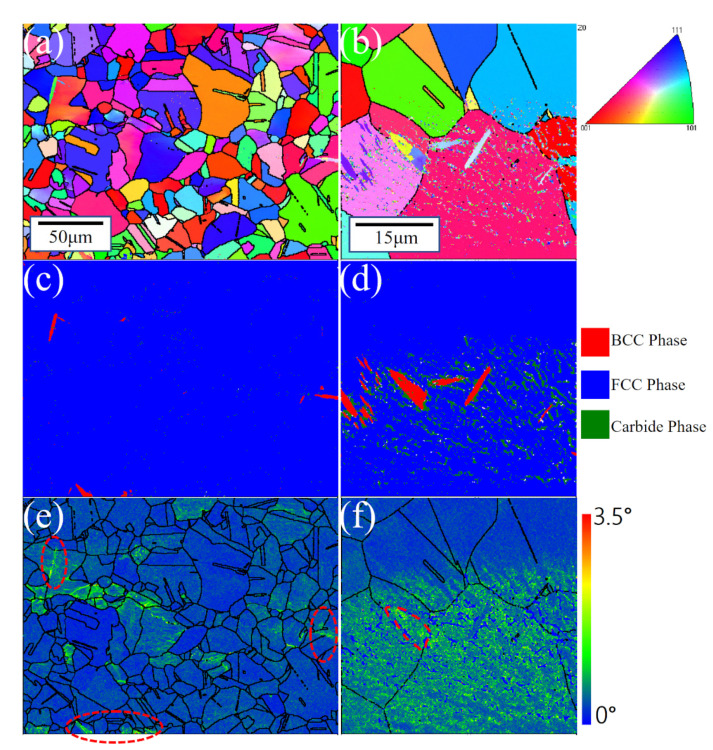
Mn test steel characterized by EBSD (**a**,**b**) is the superposition of ‘IPF’ diagram and grain boundary diagram; (**c**,**d**) is the phase distribution, where red is the BCC phase, blue is the FCC phase and green is the Carbide phase; (**e**,**f**) is the ‘KAM’ map.

**Figure 6 materials-16-02299-f006:**
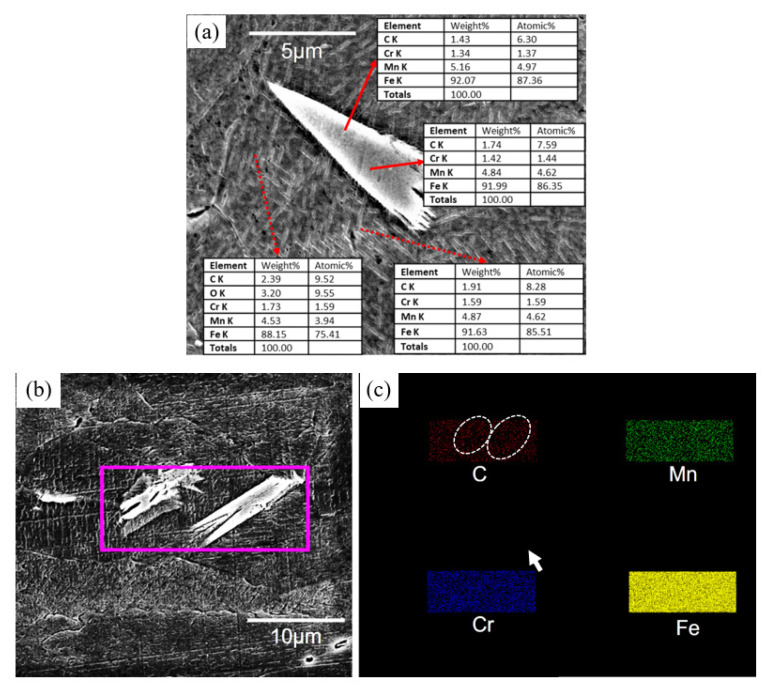
Sharp phase scan results (**a**) point energy spectrum results at base material and sharp phase; (**b**,**c**) are surface scan area and element distribution map, respectively.

**Figure 7 materials-16-02299-f007:**
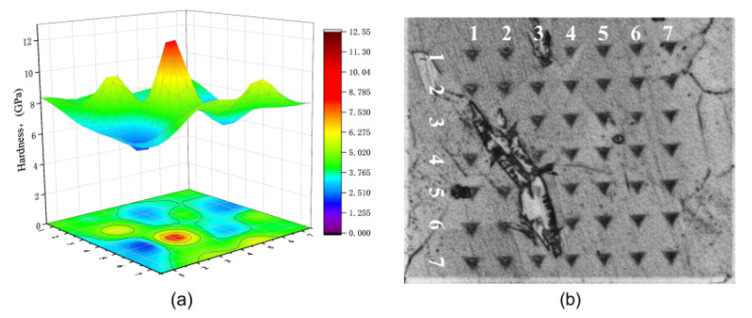
Nanoindentation results (**a**) 3D hardness figure; (**b**) metallography of indentation area.

**Figure 8 materials-16-02299-f008:**
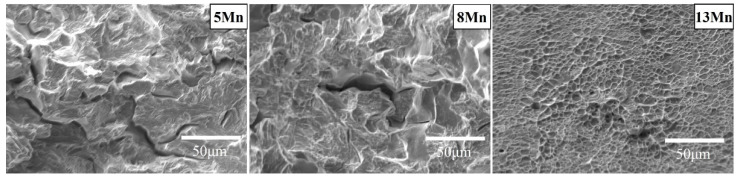
Tensile fracture morphology of different Mn content alloy at −20 °C.

**Figure 9 materials-16-02299-f009:**
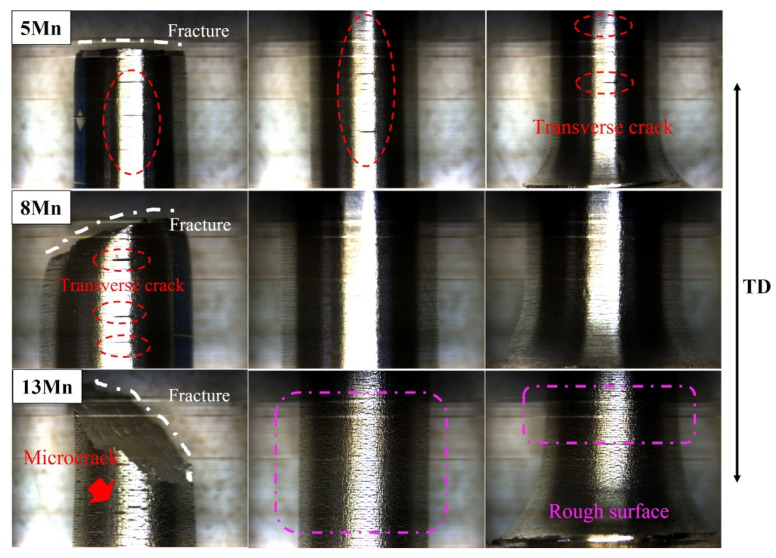
Photos of gauge length of tensile specimen (from left to right are different positions of the same sample).

**Figure 10 materials-16-02299-f010:**
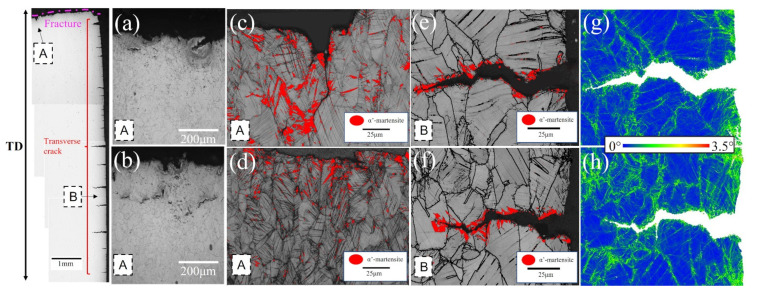
From left to right are 5 Mn tensile specimen gauge area metallographic, (**a**,**b**) fracture ‘area A’ metallographic, (**c**,**d**) ‘band contrast + α′-martensite phase distribution’ map, (**e**,**f**) gauge area edge ‘area B’ transverse crack ‘band contrast + α′-martensite phase distribution’ map and (**g**,**h**) ‘KAM’ map.

**Figure 11 materials-16-02299-f011:**
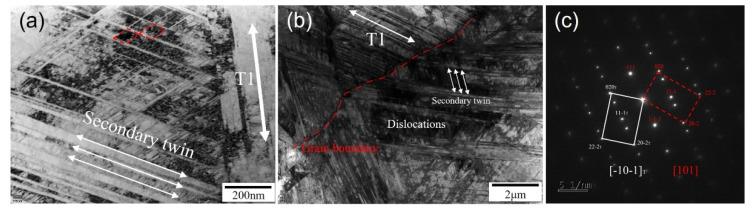
(**a**,**b**)TEM images near tensile fracture of 13 Mn test steel; (**c**) selected area electron diffraction pattern.

**Table 1 materials-16-02299-t001:** Chemical composition of experimental steels (wt/%).

Designed Steel	C	Si	V	Mn	Cr	Mo	Fe
5 Mn	1.01	0.11	0.20	5.02	1.46	0.30	Bal.
8 Mn	1.00	0.11	0.19	7.93	1.45	0.30	Bal.
13 Mn	1.02	0.10	0.20	13.22	1.47	0.29	Bal.

**Table 2 materials-16-02299-t002:** Mechanical properties and stacking fault energy of three test steels.

Steel Grade	R_m_ (MPa)	R_p0.2_ (Mpa)	A (%)	−20 °C Charpy Impact Energy (J)	SFE (mJ/m^2^)
Z1	656	580	5	9	15
Z2	864	518	23	99	21
Z3	1143	439	81	241	29

**Table 3 materials-16-02299-t003:** Phase identification in specimen tensile fracture from XRD characterization.

Compound Name	Existence	Bragg’s Angle, 2θ (hkl)	Crystal Structure	Lattice Parameter (Å)	Reference Code
α′-Martensite	5 Mn after tensile	52.3771 (110)	Body centered cubic (BCC)	a = b = c = 2.866	00-006-0696
		77.2341 (200)			
		99.7044 (211)			
γ-Austenite	5 Mn before tensile 8 Mn before/after tensile 13 Mn before/after tensile	50.7088 (111)	Face centered cubic (FCC)	a = b = c = 3.5911	00-052-0512
		59 2685 (200)			
		88.7374 (220)			
		110.1640 (311)			

## Data Availability

Not applicable.

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
