# Peer review of "Effect of Mn Content on the Toughness and Plasticity of Hot-Rolled High-Carbon Medium Manganese Steel"

_materials, 2023, doi:10.3390/ma16062299_

Round 1

Reviewer 1 Report

-           Some comments are provided below that could be helpful for the authors to improve their manuscript.

1.      The introduction is weak. The motivation for the research is missing.

2.     The author should highlight the research objectives of the research in the introduction separately. Also, how the authors would address their need to be mentioned.

3.     The manuscript lacks flow. The authors should make more efforts in presenting the work more systematically and clearly. The authors(s) have cited old citations throughout the manuscript.  The author(s) are also suggested to includes references from the latest publications (year 2022).

4.     The implications are less developed. The authors should provide more insights on it.

5.     The introduction lacks in gaining attention and highlighting the need of the study. The author(s) should mention research objectives of the research separately.

6.     The findings of the study need to be more elaborate. This section needs to be developed and supported by previous work. The discussion needs to be improvised with a theoretical contribution.The findings of the discussion need to be strengthened with the previous research work.

7.     The discussion needs to be improvised with theoretical contribution.

8.     The conclusion is very weak. It should also be an extrapolation of the key findings from the research and not a summary. So, there should be conclusions around the background theory, data theory/analysis and, key outcomes. The authors should have included the following sub-sections within the conclusion section with more details:

           Implications to theory and practice should be clearly stated;

           Key lessons learnt;

           Limitations of this research;

9.     The selection of the case location should be more elaborated.

10.  Proofread the whole manuscript as many typos and grammar errors are present.

11.   Future research directions should be improved; in that, they should stem from the awareness of the limitations and opening avenues related to the obtained outcomes

 12.  Author(s) should try to include some novel implications and unique contributions in the paper.

Good Luck

Reviewer 2 Report

The submitted communication research article on “Effect of Mn content on the toughness and plasticity of hot rolled high carbon medium high manganese steel” need to be revised properly prior to any further recommendation-

1. Please check the title. It confuses from “……hot rolled high carbon medium high manganese ….”

2. The length of the abstract needs to be shortened. Novel results most relevant to the observed phenomenon on toughness and plasticity due to varying Mn content needs to be kept.    

3. The distribution of local microstructural strain (i.e., Kernel average mis-orientation - KAM) via EBSD post processing analysis needs to be added as well in order to validate existence of different phase structure (Figure 4). Author(s) should read, and consult the following to establish –

- Microstructural investigations on simulated intercritical heat-affected zone of boron modified P91-steel

https://doi.org/10.1080/02670836.2020.1784543

- Effect of boron addition on creep strain during impression creep of P91 steel

https://doi.org/10.1007/s11665-019-04167-z

4. How Author(s) has validated analysis pertaining to XRD data as discussed in the paper? Where are the “reference codes” of the phases identified from characteristic peaks? To ease this process, Author(s) can refer following recent paper:

- Phase transformations and numerical modelling in simulated HAZ of nanostructured P91B steel for high temperature applications

https://link.springer.com/article/10.1007/s13204-018-0854-1    

*Author(s) should highlight all the modifications carried out in the paper.

Round 2

Reviewer 1 Report

Good Work

Reviewer 2 Report

Author(s) have done well. In my opinion, the paper is now ready for publication.